# Plasma Antithrombin Activity during Long-Term Magnesium Sulfate Administration for Preeclampsia without Severe Hypertension

**DOI:** 10.3390/healthcare10081581

**Published:** 2022-08-19

**Authors:** Kaori Moriuchi, Kaoru Kawasaki, Maako Hayashi, Akihiko Ueda, Yukio Yamanishi, Haruta Mogami, Kohei Fujita, Reona Shiro, Yoshie Yo, Masaki Mandai, Noriomi Matsumura

**Affiliations:** 1Department of Obstetrics and Gynecology, Faculty of Medicine, Kindai University, Osaka-Sayama 589-8511, Osaka, Japan; 2Department of Obstetrics and Gynecology, Japanese Red Cross Otsu Hospital, Otsu 520-8511, Shiga, Japan; 3Department of Gynecology and Obstetrics, Kyoto University, Kyoto 606-8507, Sakyo, Japan; 4Department of Obstetrics and Gynecology, Japanese Red Cross Wakayama Medical Center, Wakayama 640-8558, Wakayama, Japan

**Keywords:** preeclampsia, superimposed preeclampsia, magnesium sulfate, antithrombin

## Abstract

In preeclampsia, plasma antithrombin activity is decreased, which leads to exacerbation of the disorder. We previously showed that long-term magnesium sulfate (MgSO_4_) administration prolonged the pregnancy period and may be able to improve pregnancy outcomes for patients with severe preeclampsia. The present study aimed to investigate the changes in plasma antithrombin activity during long-term MgSO_4_ administration for patients without severe hypertension. This multicenter retrospective study included patients with preeclampsia and superimposed preeclampsia without severe hypertension at diagnosis. The participants were divided into two groups: MgSO_4_ nontreatment group (three institutions) and MgSO_4_ treatment group (one institution). Antithrombin activity from time of diagnosis to delivery were compared between the two groups. In the MgSO_4_ nontreatment group (*n* = 16), antithrombin activity prior to delivery was significantly lower than at time of diagnosis (*p* = 0.015). In three cases, antithrombin activity was less than 60%. On the other hand, in the MgSO_4_ treatment group (*n* = 34), antithrombin activity did not change until just before delivery (*p* = 0.74). There were no cases in which antithrombin activity was decreased below 60%. Long-term MgSO_4_ administration for preeclampsia without severe hypertension may prevent a decrease in antithrombin activity and improve the disease state of preeclampsia.

## 1. Introduction

Endothelial dysfunction is a leading cause of preeclampsia. According to the hypothesis for the pathogenesis of preeclampsia, uterine spiral artery remodeling leads to placental ischemia and hypoxia. Angiogenesis inhibitors and cytokines are then released from the ischemic and hypoxic placenta, and these cause vascular endothelial dysfunction and organ damage which can result in hypertension, proteinuria, and fetal growth restriction. Therefore, the fundamental treatment for this condition is delivery of the abnormal placenta [1]. Antihypertensive medications and anticonvulsant therapy with MgSO_4_ are just symptomatic therapies [2]. If mother and fetus are in a life-threatening situation, the pregnancy must be terminated, even in the preterm period.

To prevent and treat severe preeclampsia and eclampsia, administration of MgSO_4_ is recommended by The American college of Obstetricians and Gynecologists (ACOG), International Society for the Study of Hypertension in Pregnancy (ISSHP), Japan Society Obstetrics and Gynecology (JSOG), and Japan Association of Obstetricians and Gynecologists (JAOG) [2,3,4]. The National Institute for Health and Care Excellence (NICE) clarifies that MgSO_4_ should not be administrated for more than 24 h [5]. In threatened preterm labor, antenatal MgSO_4_ is considered for fetal neuroprotection within 24 h prior to delivery [6]. Therefore, MgSO_4_ is generally only used in short duration for a limited population of pregnant women.

Magnesium is a mineral that is essential for life, and magnesium supplementation improves endothelial dysfunction caused by cardiovascular disease [7,8]. In the condition of preeclampsia, magnesium may also ameliorate vascular endothelial dysfunction, and reduce maternal organ damage and placental dysfunction. We previously reported that MgSO_4_ administration lasting more than 48 h was able to prolong gestational periods in patients with severe early onset preeclampsia [9]. Even for patients with nonsevere preeclampsia, long-term MgSO_4_ administration may improve vascular endothelial damage and prolong the gestational period.

During pregnancy, many coagulation factors (including fibrinogen) are increased, while on the other hand, antithrombin is consumed and its level is decreased [10,11]. These changes result in hypercoagulation states during the antenatal period. In preeclampsia, endothelial dysfunction leads to microthrombosis and further consumption of antithrombin [12,13]. Additionally, the reduction in plasma antithrombin levels in preeclampsia causes further microthrombosis and vascular endothelial dysfunction. This vicious cycle increases the severity of preeclampsia.

For the past ten years at Kyoto University Hospital, MgSO_4_ has been administrated continuously to both patients experiencing preeclampsia without severe hypertension and to patients with severe hypertension, up until 24 h after delivery. In the present study, we retrospectively examined the perinatal outcomes and any changes in plasma antithrombin activity in patients with preeclampsia without severe hypertension, comparing MgSO_4_-treated and nontreated groups. This study may clarify the potential benefit of long-term MgSO_4_ treatment for preeclampsia without severe hypertension and may also provide a method of evaluation for future clinical trials.

## 2. Materials and Methods

### 2.1. Patients

A multicenter retrospective study was conducted at four institutions in Japan with the approval of the Ethical Review Board of each site (approval numbers: R03-092 Kindai University, R3123 Kyoto University, 667 Japanese Red Cross Otsu Hospital, 959 Japanese Red Cross Wakayama Medical Center).

The inclusion criteria for participants were as follows: (1) patients with preeclampsia or superimposed preeclampsia managed at each institution between January 2013 and June 2021; (2) blood pressure (BP) was in the nonsevere range at diagnosis (defined as systolic BP ≥ 140 mmHg and/or diastolic BP ≥ 90 mmHg, systolic BP < 160 mmHg and diastolic BP < 110 mmHg) [14,15]. The timing and indication for delivery were determined at each center according to the Japanese Clinical Guidelines for Obstetrical Practices [4].

### 2.2. MgSO_4_ Nontreatment Group

MgSO_4_ was not administered to patients experiencing preeclampsia without severe hypertension at Kindai University Hospital, Japanese Red Cross Otsu Hospital, or Japanese Red Cross Wakayama Medical Center, based on the Japanese Clinical Guidelines for Obstetrical practice [4]. The inclusion criteria for patient enrollment at these three institutions were as follows: (1) patients with a diagnosis of preeclampsia or superimposed preeclampsia without severe hypertension; (2) patient BP remained in the nonsevere range until delivery; (3) and plasma antithrombin activity was measured at least twice during the pregnancy.

### 2.3. MgSO_4_ Treatment Group

In Japan, the classification of hypertensive disorders of pregnancy was revised in 2018, and these changes included categorizing patients with maternal organ damage as having severe preeclampsia regardless of the severity of hypertension [15]. However, in the guidelines of ACOG and ISSHP, patients with maternal organ damage were already included in severe preeclampsia before these updates were made [2,3]. Therefore, since 2013, Kyoto University Hospital has administered continuous MgSO_4_ for all patients with preeclampsia or superimposed preeclampsia from the time of diagnosis to 24 h postpartum, regardless of the severity of hypertension. For this treatment, 4 g of MgSO_4_ was administered intravenously over 20 min, followed by continuous intravenous administration at 1 g per hour, with a maximum dose of 2 g per hour. Inclusion criteria were as follows: (1) preeclampsia or superimposed preeclampsia without severe hypertension at time of diagnosis; (2) and plasma antithrombin activity was measured at least twice during the pregnancy. Cases that led to severe hypertension before delivery were also included.

### 2.4. Evaluation Points

Clinical course and laboratory data of patients and their newborns were retrospectively reviewed via patient electronic medical records. Maternal age, gravidity, gestational weeks at diagnosis of preeclampsia or superimposed preeclampsia, MgSO_4_ administration time, maternal complications, indication for delivery, delivery mode, birth weight, fetal and neonatal death, Apgar score, and umbilical artery blood pH were measured. Additionally, data regarding antithrombin activity, platelet, fibrinogen, and uric acid levels were collected.

### 2.5. Statistical Analysis

Statistical analysis was performed using GraphPad Prism 9.3.1 (GraphPad Software, La Jolla, CA, USA). Wilcoxon signed-rank test and Pearson’s correlation were used to compare continuous variables; *p* value < 0.05 was considered statistically significant.

## 3. Results

### 3.1. MgSO_4_ Nontreatment Group

Sixteen patients were included in the MgSO_4_ nontreatment group (Table 1). Of these, 14 patients had preeclampsia and two had superimposed preeclampsia. Mean gestational weeks at diagnoses were 37.0 ± 1.8 and 33.5 ± 2.1 weeks, respectively. There were no fetal or neonatal deaths included. The mean gestational weeks of delivery was 36.9 ± 1.7 weeks, and the mean birth weight was 2486 ± 662 g. The most common indication for delivery was reaching 34 weeks of gestation (68.8%), followed by deterioration of the maternal general condition (18.8%), and deterioration of the fetal condition (12.5%).

Antithrombin activity was investigated for up to 14 days from diagnosis to delivery. Antithrombin activity just prior to delivery was significantly lower than at diagnosis (Figure 1A, Wilcoxon signed-rank test *p* = 0.015). A cutoff value of 78% antithrombin activity at the time of diagnosis of early onset preeclampsia provides a possible predictor of early delivery [16]. In this study, there were six cases (35.2%) with antithrombin activity less than 78% at diagnosis, five of which were further reduced prior to delivery. Antithrombin activity was also decreased below 60% in three cases (Figure 1B). Platelet count and fibrinogen level were positively correlated with antithrombin activity (Pearson’s correlation, r = 0.48, 0.54, *p* = 0.0049, 0.0024, respectively). On the other hand, the uric acid level was inversely correlated. (Pearson’s correlation, r = −0.36, *p* = 0.051) (Figure 2).

### 3.2. MgSO_4_ Treatment Group

Thirty-four patients were included in the MgSO_4_ treatment group (Table 2). The mean MgSO_4_ administration duration was 10.6 ± 7.5 days. There were no fetal or neonatal deaths included. The mean gestational weeks of delivery was 31.7 ± 4.0 weeks and the mean birth weight was 1389 ± 623 g. The most common indication for delivery was deterioration of the maternal general condition (44.1%), followed by deterioration of fetal condition (38.2%), and reaching 34 weeks gestation (17.6%). Three cases were complicated with HELLP syndrome.

There was no statistically significant difference in antithrombin activity at diagnosis versus just prior to delivery (Figure 3A, Wilcoxon signed-rank test *p* = 0.74). In 12 cases, antithrombin activity was decreased below 78% at the beginning of MgSO_4_ administration. There were no cases in which antithrombin activity was decreased below 60% (Figure 3B). Platelet count and fibrinogen level were positively correlated with antithrombin activity (Pearson’s correlation, r = 0.25, 0.22, *p* = 0.0057, 0.015, respectively). On the other hand, the uric acid level was negatively correlated. (Pearson’s correlation, r = −0.29, *p* = 0.0026).

## 4. Discussion

This study investigated antithrombin activity with and without MgSO_4_ administration in patients experiencing preeclampsia without severe hypertension. In the MgSO_4_ treatment group, the majority of participants presented with early onset cases and were diagnosed with preeclampsia at less than 30 weeks of gestation (Table 2). Early onset preeclampsia is assumed to be caused by placental dysplasia, and even if the patients have nonsevere hypertension at the time of diagnosis, the condition often worsens until delivery [17]. Internationally, this research is the first to suggest that MgSO_4_ may be involved in vascular endothelial protection for such cases.

The Food and Drug Administration states that MgSO_4_ should not be used for more than 5–7 days to prevent preterm birth, which is due to concerns about the effects on children [18]. This recommendation is based on case reports about bone loss and fractures in infants who were administrated MgSO_4_ for more than 7 days antenatally. However, recently a large population-based cohort study reported that long-term MgSO_4_ administration during pregnancy did not actually increase the risk of bone fractures in infants [19]. Therefore, because the safety of long-term MgSO_4_ administration has been confirmed, its usefulness should be reevaluated.

Magnesium significantly improves vascular endothelial function in the cardiovascular system [20], and low magnesium levels generate free radicals and increase inflammatory cytokines which induce vascular permeability. The latter situation leads to atherosclerosis, thrombus formation, and hypertension [7,21]. Epidemiologically, magnesium in serum, plasma, and diet is inversely associated with risk factors for cardiovascular disease [8]. We previously reported that MgSO_4_ administration decreased urinary oxidative stress marker 8-isoprostane in severe preeclampsia [22]. Based on these findings, magnesium may also improve endothelial oxidative stress and reduce vascular endothelial dysfunction in the condition of preeclampsia.

In preeclampsia, endothelial dysfunction leads to microthrombus formation and decreases antithrombin levels and activity [12,13], and low antithrombin activity is a predictor of early delivery for patients with preeclampsia [16]. In the present study, antithrombin activity was significantly correlated not only with factors involved in coagulation, but also with uric acid levels (Figure 4). These results suggests that antithrombin activity is an important indicator of the severity of preeclampsia. Additionally, even in MgSO_4_ nontreated preeclampsia without severe hypertension, antithrombin activity was significantly reduced, as previously reported [13] (Figure 1). This suggests that a microthrombus is formed toward the end of the pregnancy. On the other hand, in the MgSO_4_ treatment group, antithrombin activity was not decreased below 60% in any of the cases. MgSO_4_ may maintain antithrombin activity, which normally declines in pregnancy (Figure 3).

Antithrombin exhibits both anticoagulant and anti-inflammatory effects. Antithrombin binds to a membrane-bound portion of the vascular endothelial cell surface called syndecans, thereby promoting prostacyclin production and suppressing the inflammatory response in the vascular endothelium [23]. In a rat sepsis model, antithrombin was shown to protect the endothelial glycocalyx and maintain vascular integrity [24]. In acute respiratory distress syndrome caused by vascular endothelial dysfunction, recombinant antithrombin supplementation preserves the endothelial glycocalyx and ameliorates the condition [25]. In preeclampsia, antithrombin supplementation suppresses microthrombosis, protects the vascular endothelium, and reduces inflammation, suggesting that antithrombin supplementation may improve prognosis [26,27]. A phase 3 study is currently underway to evaluate the effects of antithrombin replacement on patients with preeclampsia [28]. Long-term MgSO_4_ administration may also maintain plasma antithrombin activity, with similar effects to antithrombin replacement therapy.

There were several limitations to the present study. First, this is a retrospective study that include a relatively small number of patients. Second, the patients in the MgSO_4_ nontreated group were less severe than in the MgSO_4_-treated group, because the nontreated group consisted of patients who remained in nonsevere hypertension until delivery and did not require MgSO_4_ administration. Third, this research is a pilot study because there is case bias. We intend to conduct a prospective study in the future, matching the conditions of the treated and nontreated groups. Prospective randomized clinical trials are essential to determine the safety and benefit of long-term MgSO_4_ administration for patients experiencing preeclampsia without severe hypertension at diagnosis.

In conclusion, long-term MgSO_4_ administration for patients experiencing preeclampsia without severe hypertension at diagnosis may prevent the decline in antithrombin activity, delay disease progression, and improve patient prognosis.

## Figures and Tables

**Figure 1 healthcare-10-01581-f001:**
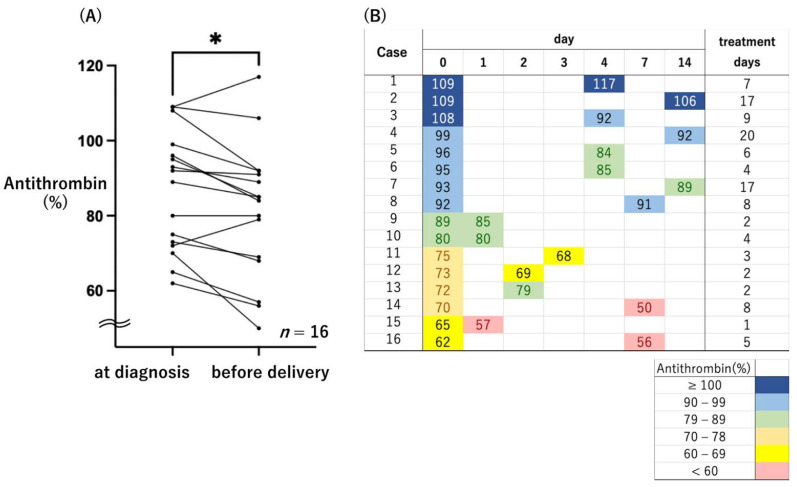
Plasma antithrombin activity in the MgSO_4_ nontreatment group. (**A**) Plasma antithrombin activity at diagnosis and just before delivery. * *p* = 0.015 (**B**) Transition of plasma antithrombin activity from diagnosis to delivery. Treatment days means number of days from diagnosis to delivery. The degree of antithrombin activity was identified by the six colors shown in this figure (Navy: ≥100%; Blue: 90–99%; Green: 79–89%; Orange: 70–78%; Yellow: 60–69%; Red: <60%).

**Figure 2 healthcare-10-01581-f002:**
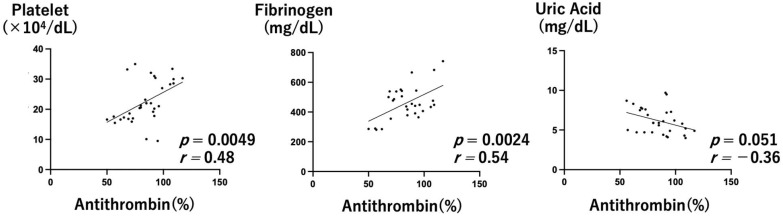
Correlations between antithrombin activity with other factors (platelet, fibrinogen, and uric acid) in the MgSO_4_ nontreatment group.

**Figure 3 healthcare-10-01581-f003:**
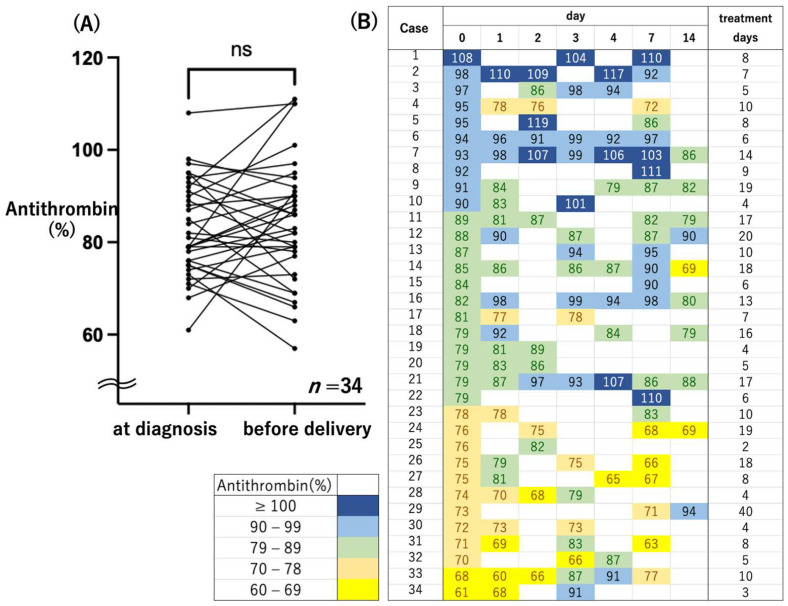
Plasma Antithrombin activity in the MgSO_4_ treatment group. (**A**) Plasma antithrombin activity at diagnosis and just before delivery. ns, nonsignificant; *p* = 0.74 (**B**) Transition of plasma antithrombin activity from diagnosis to delivery. Treatment days means number of days from diagnosis to delivery. The degree of antithrombin activity was identified by the six colors shown in this figure (Navy: ≥100%; Blue: 90–99%; Green: 79–89%; Orange: 70–78%; Yellow: 60–69%).

**Figure 4 healthcare-10-01581-f004:**
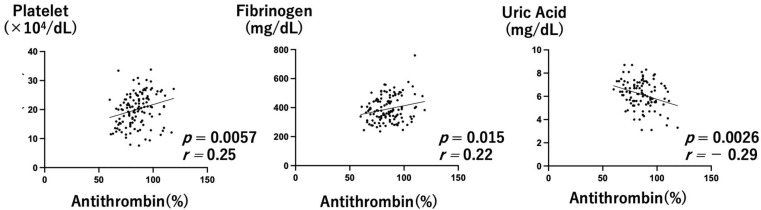
Correlations between antithrombin activity and other factors (platelet, fibrinogen, and uric acid) in the MgSO_4_ treatment group.

**Table 1 healthcare-10-01581-t001:** Patient characteristics and obstetric and perinatal outcomes in the MgSO_4_ nontreatment group. PE: preeclampsia; SPE: superimposed preeclampsia.

	*n* = 16
**Patient characteristics**	
Age (years, mean ± SD)	30.6 ± 5.9
Nullipara (%)	15 [93.8%]
Onset of PE or SPE (weeks, mean ± SD)	35.8 ± 2.6
Number of PE (*n*, (%))	14 [87.5%]
Onset of PE (weeks, mean ± SD)	37 ± 1.8
Number of SPE (*n*, (%))	2 [12.5%]
Onset of SPE (weeks, mean ± SD)	33.5 ± 2.1
**Obstetric outcomes**	
Cesarean delivery (%)	50%
Gestational weeks of delivery (weeks, mean ± SD)	36.9 ± 1.7
Indication for delivery (*n*, (%))	
Fetal	2 [12.5%]
Maternal	3 [18.8%]
Attainment of 34–37 weeks’ gestation	11 [68.8%]
Major maternal complications (*n*, (%))	
Placental abruption	0
HELLP syndrome	0
**Neonatal outcomes**	
Neonatal and fetal death (%)	0
Birth weight (g, mean ± SD)	2486 ± 662
1 min Apgar score	7.8 ± 1.2
5 min Apgar score	8.8 ± 1.0
Cord pH	7.27 ± 0.1

**Table 2 healthcare-10-01581-t002:** Patient characteristics and obstetric and perinatal outcomes in the MgSO_4_ treatment group. PE: preeclampsia; SPE: superimposed preeclampsia.

	*n* = 34
**Patient characteristics**	
Age (years, mean ± SD)	34.8 ± 5.7
Nullipara (%)	27 [79.4%]
Onset of PE or SPE (weeks, mean ± SD)	29 ± 4.7
Number of PE (*n*, (%))	29 [85.3%]
Onset of PE (weeks, mean ± SD)	29.7 ± 4.4
Number of SPE (*n*, (%))	5 [14.7%]
Onset of SPE (weeks, mean ± SD)	24.8 ± 5.1
Duration of MgSO_4_ use (days, mean ± SD)	10.6 ± 7.5
**Obstetric outcomes**	
Cesarean delivery (%)	73.5
Gestational weeks of delivery (weeks, mean ± SD)	31.7 ± 4.0
Indication for delivery (*n*, (%))	
Fetal	13 [38.2%]
Maternal	15 [44.1%]
Attainment of 34–37 weeks’ gestation	6 [17.6%]
Major maternal complications (*n*, (%))	
Placental abruption	0 [0%]
HELLP syndrome	3 [8.8%]
**Perinatal outcomes**	
Neonatal and fetal death (%)	0
Birth weight (g, mean ± SD)	1389 ± 623
1 min Apgar score	5.9 ± 2.7
5 min Apgar score	7.7 ± 1.9
Cord pH	7.26 ± 0.1

## Data Availability

The data that support the findings of this study are available from the corresponding author.

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
