# Peer review of "Plasma Antithrombin Activity during Long-Term Magnesium Sulfate Administration for Preeclampsia without Severe Hypertension"

_healthcare, 2022, doi:10.3390/healthcare10081581_

Round 1

Reviewer 1 Report

This retrospective review of records tested the hypothesis that magnesium sulfate administration for preeclampsia is associated with higher anti-thrombin activity.  It is generally well-done and well-written, and most of my comments are about fairly minor issues.  However, the English used is stilted and unnatural in places and should be reviewed by a native English speaker.

Abstract

Third sentence:  Without the context of an explanation of your method, the term “patients without severe preeclampsia” is confusing.  Can you use the term you used in your title, “preeclampsia without severe hypertension”?

Introduction

Well done.

Materials and Methods

Please explain why the institutions in the non-treatment group do not administer MgSO4 to their preeclampsia patients.  If it is because of the government recommendations you mentioned in your introduction, please state that, but then explain why Kyoto University Hospital would go against those recommendations and routinely use MgSO4.  

Please state the diagnostic criteria for both preeclampsia and superimposed preeclampsia without severe hypertension and cite references.  You gave the upper limits of the blood pressure range for non-severe, but please also include the lower limits, since you are still talking about a hypertensive patient.

Discussion

Your conclusions about cause and effect of MgSO4 administration and pregnancy outcome could perhaps be softened, since your control group was less severe than the treatment group.  You might even characterize your research as a “pilot study”, restate your found associations, then can state that you suspect that MgSO4 causes a certain outcome after summarizing observations of associations.

Figure 3

The information presented is very interesting, but in 3A the sixteen lines are so close together that they are nearly impossible to make out.  It might be helpful to show only one line, which is an average of the individual results.

References

Current, scholarly, and sufficient in number

Reviewer 2 Report

 I am very excited to be able to review this study. It has a meaningful impact on maternal and childrens health and should be published recently. The title of the study is clear, shows the type of the study, and is well constructed. As well as the whole article.

In Introduction section authors presented perfect what is known in the literature and the gap in the knowledge. The aim of the study could be more underlined but it is presented in appropriate place.

The STROBE guidelines checklist should be fulfilled and added to the study as the supplementary file, but all needed information are presented in the study

One thing remain unclear. When did you dicided to give the treatment group Magnesium. What was the indication? Every patient included to tretment group with preeclampsia without severe hypertension by admision have got Magnesium iv? Am I good andesenting the inclusion to the study?

I have missed also in your article the everage / mediane time MgSO4 was administred. The Spearman correlation of MgSO4 dosis / time of administration with Antitrombine continued measurement would be interesting to see.
